# Perspectives and experiences of Covid-19: Two Irish studies of families in disadvantaged communities

**Catarina Leitão** ⓘ *, **Jefrey Shumba, Marian Quinn**

Childhood Development Initiative, Dublin, Republic of Ireland

* catarina@cdi.ie, catarinafcl@gmail.com

**Data Availability Statement:** Qualitative data files are available from the Zenodo database (DOI: https://doi.org/10.5281/zenodo.6647761). Due to ethical restrictions related to protecting participant privacy set by The Childhood Development

## Abstract

The Covid-19 pandemic has been recognised to affect families' socio-emotional well-being. Collecting the views of families in diverse socio-economic contexts can contribute to understanding their specific needs and resources in relation to the Covid-19 pandemic. The overarching objective of the current research was to explore the views and experiences of families in relation to the Covid-19 pandemic, who were living in the Republic of Ireland, including in areas designated as disadvantaged. In Study 1, the objective was to explore changes, difficulties, and concerns experienced by parents of children up to six years old during the pandemic, and related associations with socio-demographic characteristics. Data were collected from 168 parents/carers via an online questionnaire, and examined using conceptual content analysis. The most frequently identified experiences related to restrictions, social isolation, negative impacts on parents' emotional and psychological well-being, negative impacts on children's emotional well-being and development, concerns with physical health, uncertainty about the future, and positive changes regarding family time and activities. Associations were found with parents' age and work situation, and family's income and composition. In Study 2, the objective was to explore the views of children, parents, and service providers about the impact of the Covid-19 pandemic on families' life, and relevant supports. Data were collected from 50 children aged between eight and 17 years old, 17 parents, and 20 service providers through focus group discussions, and examined using thematic analysis. The participants reported experiences related to challenges with online education, uncertainty regarding children's education, food poverty, and children's socio-emotional health. The findings of both studies reinforced the importance of implementing measures to promote parents' and children's socio-emotional well-being, combat educational inequalities, and ensure economic and employment security.

## Introduction

The Coronavirus disease (Covid-19) was characterised as a pandemic by the World Health Organisation on the 11th of March 2020. In the same month, nationwide measures were implemented in the Republic of Ireland to avoid the spreading of the virus. These measures included

Initiative (CDI), the full dataset with demographic information used in Study 1 cannot be made publicly available. For more information, please contact CDI via info@cdi.ie.

**Funding:** Study 1 was conducted within a project that has received funding from the European Union's Horizon 2020 research and innovation programme under the Marie Skłodowska-Curie grant agreement No 890925 (https://ec.europa.eu/programmes/horizon2020/en/tags/horizon-2020-research-and-innovation-programme). Funding was awarded to CL. Study 2 was funded by Tusla under the Area Based Childhood funding and the Child and Youth Participation Initiatives grant (https://www.ncirl.ie/Area-Based-Childhood-Programme). Funding was awarded to JS. The funders had no role in study design, data collection and analysis, decision to publish, or preparation of the manuscript.

**Competing interests:** Study 1 was conducted within a project that has received funding from the European Union's Horizon 2020 research and innovation programme under the Marie Skłodowska-Curie grant agreement No 890925. Study 2 was funded by Tusla under the Area Based Childhood funding and the Child and Youth Participation Initiatives grant. This does not alter our adherence to PLOS ONE policies on sharing data and materials

staying at home for the general population, closure of schools, crèches, and other childcare facilities, and remote working where possible, among others. The Republic of Ireland had one of the longest closure of public spaces in Europe during the first wave of the pandemic [1], and reintroduced a partial lockdown during the second wave. Prolonged and substantial restrictions on public life can have a negative impact on individuals' and families' lives, which can persist after they are lifted [2, 3].

The Covid-19 pandemic led to the loss of human lives, health and well-being impacts, and socio-economic changes. Individuals and families have faced uncertainty and insecurity, and many have experienced illness and bereavement. A higher risk of Covid-19 infection (suspected/confirmed cases, living in hard-hit areas, having pre-existing health conditions) can also increase the likelihood of anxiety and depression [4].

The Covid-19 containment and mitigation measures have particularly affected families. Research has demonstrated lockdown-related behaviours, including anxiety and stress in parents and children, and behavioural problems in children [5–7]. Parents identified social distancing, closure of childcare services and schools, and job-related changes as challenging in previous studies [7–9]. Parents of children not attending childcare services because of Covid-19 reported feeling stressed, worried or overwhelmed, particularly mothers, as well as noting that the closures had a negative impact on their child's social and emotional development and well-being [10]. One Irish study found that parents described the closure of childcare services as impacting on children through increases in tantrums, anxiety, clinginess, boredom and under-stimulation [11].

Losing access to childcare services and schools can particularly affect children from disadvantaged socio-economic backgrounds due to differentiated access to learning resources at home [12]. In previous research, early education providers indicated that not accessing childcare services can particularly and negatively impact the physical development of children from deprived homes [10]. Primary school staff were also found to perceive an increase in inequalities and gaps in learning, health, and well-being during school closures [13].

The challenges brought about by the pandemic and its associated restrictions can be more accentuated for those living in low-income households, and for those at risk of social exclusion and poverty prior to the pandemic [14]. Although the Republic of Ireland was the only Member State of the European Union (EU) to register positive GDP growth in 2020, its domestic economy still felt the impact of pandemic restrictions in early 2021 [15]. Employment rates fell in almost all EU Member States between 2019 and 2020, including in the Republic of Ireland, and even more for foreign-born people [14]. Individuals with lower income before the pandemic were more likely to experience further income losses [16]. Large families (two adults with three or more dependent children) and single-parent families were identified as being at higher risk of severe housing deprivation (i.e. overcrowded dwelling, leaking roof, no bath/shower, no indoor toilet, or a dark dwelling), which can present increased challenges when required to stay at home [14].

During the pandemic, people with a paid job, living with a partner, with better general health, exercising daily, and avoiding loneliness were found to report less dissatisfaction and anxiety [9]. Another study indicated that the less adequate the physical space where people were isolated, and the longer the isolation, the worse the mental health [17]. Covid-19 impacts associated with poorer families' mental health were found to include self-quarantine, financial hardship, and family stress [18].

There is a growing body of research focused on the experiences of parents during the pandemic, and on their views about children's experiences [5, 6]. However, collecting the views and experiences of families in diverse socio-economic contexts is necessary to better understand their needs and resources. More particularly, understanding the socio-emotional impact

of the pandemic and related restrictions on young children was identified as critical [11]. Collecting the views from service providers working with families can also contribute to better understand how their needs and resources can be effectively supported [13]. These insights can inform future practices and policies to support families in addressing the short and long-term effects of Covid-19.

The overarching objective of the current exploratory research was to understand the views and experiences of families during the Covid-19 pandemic, most of whom were living in areas of Dublin designated as disadvantaged using the Pobal HP Deprivation Index [19]. In Study 1, we analysed the changes, difficulties and concerns of parents and carers of children aged up to six years old in relation to the pandemic, and related associations with socio-demographic characteristics. In Study 2, we examined the views and experiences of children (eight-17 years old), parents with children aged under 18 years, and service providers working with families about the impact of the pandemic on families' life, and relevant supports.

Studies 1 and 2 occurred near the end of the pandemic's second wave and the beginning of the third wave. Presenting the findings together was seen as the most effective way to provide a comprehensive picture of the experiences of families with children of different ages living in areas with a similar deprivation profile. Study 1 used quantitative analysis to explore the range and frequency of families' experiences, and associations with socio-demographic characteristics. Study 2 used a qualitative methodology to explore the depth of the experiences of children and families during the pandemic. The two studies considered together can better contribute to offering insights into the effects of the pandemic on families living in disadvantaged communities.

## Study 1

Study 1 had two specific objectives. The first objective was to explore the views of parents and carers of young children (up to six years old) about the changes experienced with the Covid-19 pandemic, difficulties regarding the restrictions, and current and future concerns about Covid-19. The second objective was to explore what socio-demographic characteristics of the parents and carers were associated with experienced Covid-19-related changes, difficulties, and concerns.

### Methodology

**Participants.** There were 168 participants in this study, all of whom were parents or carers of at least one child in the birth to six years age range, and most of whom were living in designated disadvantaged areas of Dublin (according to the Pobal HP Deprivation Index). In total, 157 (93%) identified themselves as women, and 11 (7%) as men; 153 (91%) were mothers, 10 (6%) were fathers, and five (3%) were other carers of children in the specified age range; 137 (82%) were born in Ireland and 31 (18%) in another country. The mean age of the participants was 34.34 years (SD = 6.11). Among the participants, 62 (37%) had an educational level between 6 and 8 according to the International Standard Classification of Education (ISCED), corresponding to tertiary education in the Republic of Ireland; 77 (46%) had a level between ISCED 3 and 5 (secondary education); and 23 (14%) had a level of ISCED 2 (intermediate education) or below. In total, 97 (58%) were in a paid employment (full or part-time), and 71 (42%) were not in a paid employment; 86 (51%) were not receiving any social welfare payments, and 77 (46%) were receiving at least one type of social welfare payment; 83 (49%) did not have a medical card, 19 (13%) had a medical card for a General Practitioner only, and 63 (38%) had a full medical card. In the Republic of Ireland, to qualify for a medical card, the weekly income must be below a certain amount for the family size. The income limits for the

General Practitioner only card are higher than the limits for the full medical card [20]. Participating households had an average of two children (M = 2.00; SD = 1.04), and two adults (M = 2.06; SD = 0.80).

**Instruments.** Socio-demographic data were collected on parents'/carers' gender (woman, man, or other); age; country of birth (Republic of Ireland or other); highest educational attainment level completed; work situation; medical card (no coverage, General Practitioner only, or full coverage); and receipt of social welfare payments. The questions on educational attainment level and work situation were adapted from the 2016 Census of Population of Ireland [21]. Parents/carers were asked about their relationship to the children attending the Early Learning and Care service from which they received the invitation to participate, and the children's age. Parents/carers were also asked to indicate the number of children and adults in their household.

Data on the experiences related to Covid-19 were collected through three open-ended questions, namely: What changes did you experience with the Covid-19 pandemic? What did you find difficult about the Covid-19 restrictions? What are your current and future concerns regarding Covid-19?

Data were collected through an online questionnaire, which was pilot tested.

**Data collection.** The research team used purposive sampling to recruit participants, by contacting Early Learning and Care services located in areas defined as disadvantaged according to the Pobal HP Deprivation Index. The researchers asked the Early Learning and Care services to provide parents and carers with information on the study, and to invite them to participate (by sharing the information sheet on the study and the link to the online questionnaire). Given that the questionnaire was accessible to anyone with the internet link, the research team recognised that snowball sampling may also have taken place. A relationship between the research team and the participants was not established prior to study commencement.

Data were collected through an online questionnaire, which was developed within a study that had the objective of evaluating the impact of a parenting support programme. More questions were asked in addition to those analysed here. The whole questionnaire was expected to take up to 20 minutes to be completed.

Data were collected between December 2020 and January 2021. In this period, the Republic of Ireland was entering the third wave of the Covid-19 pandemic [22].

**Ethical considerations.** Ethics approval for the study was obtained from the national Child and Family Agency's (Tusla) Research and Ethics Committee prior to commencing the study. Information on the study and a consent form were included at the start of the online questionnaire. The information on the study included the reasons for the research, data treatment procedures, and researcher's role and contact. Participants were required to provide informed consent before continuing to the questionnaire by clicking checkboxes on the screen. Data were treated confidentially and anonymised.

**Reflexivity.** The author who developed the online questionnaire and was involved in the data analysis (CL) was a female Research Fellow with a PhD in Social Psychology. The second author involved in the data analysis (JS) was a male data specialist with a Masters in Applied Social Research. The authors had previous training and experience in collecting and analysing qualitative data focused on the views of parents and carers of children. While the authors recognised that personal and contextual aspects could shape the research, they endeavoured to remain grounded in the data and avoid imposing personal assumptions about families' experiences on the analysis.

**Data analysis.** Participants' responses were exported to Microsoft Excel. A dataset was created for each of the three questions, namely on changes, difficulties, and concerns related to the Covid-19 pandemic. The three datasets were analysed separately.

Data were analysed using conceptual content analysis. All the responses were read multiple times to get an overview of the data. Notes were added next to each response to condense the meaning units. The meaning units could correspond to words, expressions, sentences, or the whole response. Notes with similar content were grouped for the development of codes. For each code, a description and a name were generated based on the data, and examples were included. A coding scheme was created. Participants whose responses fitted into a code were assigned a 1 for that code; otherwise, a 0. Participants could be assigned a 1 for more than one code when their responses included different meaning units. Missing responses were not included in the analysis.

Approximately 20% of the participants (n = 40) were randomly selected within each dataset, and their responses were coded separately by the first two authors, using the coding scheme. The scores given by the researchers were exported to IBM SPSS version 28. Interrater reliability was measured by calculating the Cohen's kappa [23]. After the analysis of the interrater reliability, all participants were assigned a score for each code.

Based on the description of each code, codes were grouped into higher order categories. Participants whose response fitted into at least one of the codes included in a category were assigned a 1 for that category; otherwise, a 0.

To explore what socio-demographic characteristics were associated with participants' changes, difficulties, and concerns according to the categories created, we conducted logistic regression analyses.

The results were not shared with the participants to be able to provide feedback, given that during data collection their consent to be contacted for this purpose was not collected.

## Results

With the objective of exploring changes, difficulties, and concerns related to Covid-19 experienced by parents and carers of young children, codes and categories were identified through qualitative analysis. The interrater reliability was high at code level (k≥.70). In total, 18 codes and 11 categories were identified in relation to changes experienced due to the pandemic (Table 1); 20 codes and 10 categories were identified regarding difficulties with restrictions (Table 2); and 19 codes and nine categories were identified within current and future concerns (Table 3). The majority of codes and categories identified were common across the three themes.

The following categories were identified within changes, difficulties, and concerns: Family social isolation (i.e. less interaction with others); Negative impact on children's emotional well-being and development; Structural restrictions (i.e. mitigation and containment measures implemented by the government); Decreased stability of material resources (related to unemployment, decreased financial capability, and finding or keeping a house); Physical health concerns; Positive changes in family's time spent together and activities; and Absence of changes, difficulties or concerns (when the participant indicated there were none or almost none). Categories which were identified in both changes and difficulties were Negative impact on parents/carers' emotional and psychological well-being, and Changes in the family routine. Two categories were identified only in relation to changes: Changes in the family constitution (which included having a child during the pandemic, a separation, or loss of family members), and Generalised changes (when the participant indicated that a lot had changed, without further details). The category Uncertainty about the future was identified in the concerns data only.

Within the changes experienced, the most frequent category referred to Structural restrictions, with 50% of the participants reporting changes related to one or more restrictions (closure or limited access to childcare services/schools, changes in working conditions, and/or

**Table 1. Codes and categories within changes related to the Covid-19 pandemic.**

| Category and codes | Example | n | % |
|---|---|---|---|
| Social isolation | | 44 | 26 |
| Social isolation of parents or family in general | *Not visiting extended family as much as before.* | 40 | 24 |
| Social isolation of children | *The children missed their friends more.* | 14 | 8 |
| Negative changes in parents' emotional and psychological well-being | | 52 | 31 |
| Negative changes in parents' emotional well-being | *Mental health issues I used to have were back.* | 45 | 27 |
| Increased demands on parents | *The balance of working full time remotely as well as childcare facilities closing meant that I had less time to give to my child and less time to dedicate to my work.* | 19 | 11 |
| Negative changes in children's emotional well-being and development | | 11 | 7 |
| Negative changes in children's emotional well-being | *Anxiety in both myself and my children worsened.* | 9 | 5 |
| Negative changes in children's development | *Less exploration of the world and children need to be exposed for sensory development, however due to Covid we couldn't do that.* | 3 | 2 |
| Structural restrictions | | 85 | 50 |
| Closure or limited access to childcare services/schools | *No childcare.* | 34 | 20 |
| Changes in working conditions | *Working from home.* | 35 | 21 |
| Decreased time or activity outside the home | *Staying at home a lot more.* | 39 | 23 |
| Decreased stability of material resources | | 21 | 13 |
| Unemployment | *I lost my job.* | 17 | 10 |
| Decreased financial capacity | *Financial difficulties.* | 4 | 2 |
| Positive changes in personal and family life | | 33 | 20 |
| Positive changes in family time and activities | *I learned how to enjoy family time more and appreciate close family.* | 29 | 17 |
| Positive changes in personal time and activities | *Enjoying working from home. Built a home gym and get to train every day.* | 5 | 3 |
| Physical health concerns | *Lots of hand sanitizer in our home along with temperature checks.* | 9 | 5 |
| Changes in family constitution | *Loss of a sibling.* | 8 | 5 |
| Changes in routine | *No routine for kids not in school.* | 3 | 2 |
| General changes not detailed | *Everything has changed and we are missing the normal days.* | 7 | 4 |
| Absence of changes | *Not a lot.* | 6 | 4 |

having less time and fewer activities outside the home). Negative changes in parents' emotional and psychological well-being, and Social isolation were the second and third most frequent categories within changes, being experienced by 31% and 26% of the participants, respectively. The fourth category most frequently identified was Positive changes regarding family's or personal time and activities, with this being reported by 20% of the participants.

The most frequent category identified in relation to difficulties was Social isolation, being reported by 56% of the participants, followed by Structural restrictions, with 49% of the participants reporting difficulties with one or more restrictions (most frequently insufficient time or activity outside the home). The third and fourth categories most frequently addressed were Negative impact on parents' emotional and psychological well-being, and Children's

**Table 2. Codes and categories within difficulties related to the Covid-19 restrictions.**

| Category and code | Example | n | % |
|---|---|---|---|
| Social isolation | | 94 | 56 |
| Social isolation of parents or family in general | *Having to stay away from all our loved ones and friends was extremely tough and heart-breaking. As a family we'd be very sociable so that was a struggle.* | 81 | 48 |
| Social isolation of children | *Keeping my child away from family which he loves to visit.* | 17 | 10 |
| Negative impact on parents' emotional and psychological well-being | | 34 | 20 |
| Negative impact on parents' emotional well-being | *I feel it had a real bad impact on us me and my child.* | 21 | 13 |
| Negative impact of increased demands on parents | *No break. Trying to work with children at home.* | 17 | 10 |
| Negative impact on children's emotional well-being and development | | 25 | 15 |
| Negative impact on children's emotional well-being | *Being stuck at home when kid's would cry to be out.* | 22 | 13 |
| Negative impact on children's development | *My child's development deteriorating due to school closures.* | 3 | 2 |
| Structural restrictions | | 83 | 49 |
| Closure or limited access to childcare services/schools | *Especially closing down of schools.* | 20 | 12 |
| Difficulties related to working conditions | *Working remotely.* | 9 | 5 |
| Limited access to services (other than childcare services/schools) | *Not being able to access services (hospital/specialist appointments).* | 4 | 2 |
| Insufficient time or activity outside the home | *Not having enough outside time and not being able to do different things with the kids.* | 53 | 32 |
| Mask use | *Wearing masks though it really is important, sometimes I find it very uncomfortable and difficult to breathe.* | 5 | 3 |
| Difficulties with general restrictions or lockdowns | *Being so restricted.* | 6 | 4 |
| Lack of clarity of the restrictions | *I found it quite difficult that the restrictions changed and weren't very clear.* | 1 | 1 |
| Decreased stability of material resources | | 6 | 4 |
| Unemployment | *Not working.* | 1 | 1 |
| Decreased financial capacity | *Financially we had to tighten our belts and prioritise the bills.* | 4 | 2 |
| Difficulties with housing | *Finding a home.* | 1 | 1 |
| Physical health concerns | *My child's eating is so bad at home compared to school.* | 3 | 2 |
| Positive changes in family life | *It's been a big change but mostly positive, as I am really enjoying the time with my child.* | 1 | 1 |
| Difficulties with changes in the routine | *Routine being badly affected. Not being able to get into a good routine for my son.* | 6 | 4 |
| Absence of difficulties | *Nothing.* | 4 | 2 |

emotional well-being and development, being reported by 20% and 15% of the participants, respectively.

Within current and future concerns, the category most frequently identified was Physical health, cited by 30% of participants. The second, third and fourth categories most frequently addressed were Uncertainty about the future, Continuity of structural restrictions (most frequently the closure of childcare services/schools, and general lockdowns), and Social isolation, which were mentioned by 23%, 22%, and 18% of the participants, respectively.

**Table 3. Codes and categories within concerns related to the Covid-19 pandemic.**

| Category and Code | Example | n | % |
|---|---|---|---|
| Social isolation | | 31 | 18 |
| Social isolation of parents or family in general | *Less interaction with family members.* | 25 | 15 |
| Social isolation of children | *Children not seeing their grandparents* | 8 | 5 |
| Negative impact on children's emotional well-being and development | | 27 | 16 |
| Negative impact with children's emotional well-being | *Will the loss of affection affect my child? (ie not hugging friends/ family, shaking hands when meeting new people).* | 19 | 11 |
| Negative impact on children's development | *Future concerns are the impact on the kids being out of school so much.* | 14 | 8 |
| Continuity of structural restrictions | | 37 | 22 |
| Childcare/schools closure | *The uncertainty of whether schools/crèches can be closed at a moment's notice.* | 17 | 10 |
| Concerns related to working conditions | *Prospect of having to work from home again.* | 5 | 3 |
| Closure of services (other than childcare services/schools) | *Not getting enough services to open.* | 3 | 2 |
| Insufficient activity outside the home | *Not able to take them [children] to new places like beaches because they are too far.* | 5 | 3 |
| General restrictions | *More lockdowns.* | 13 | 8 |
| Decreased stability of material resources | | 18 | 11 |
| Unemployment | *Job loss.* | 10 | 6 |
| Decreased financial capacity | *Being unable to provide financially for my children.* | 10 | 6 |
| Difficulties with housing | *That my husband will remain out of work and we are at risk of losing our home.* | 2 | 1 |
| Physical health concerns | | 51 | 30 |
| Negative impact on physical health | *My current concern is that Covid is highly infectious and it might infect my child.* | 36 | 21 |
| Concerns about the use and effectiveness of the vaccine | *Not enough people will take vaccine.* | 18 | 11 |
| Societal concerns (beyond personal or family life) | *The mental health effect this will have on so many people.* | 19 | 11 |
| Uncertainty about the future | *That the world will never return to what it was.* | 39 | 23 |
| Positive changes in family life | *To make time for family more. It's more important than work.* | 1 | 1 |
| Absence of concerns | | 19 | 11 |
| Indefiniteness of concerns | *Really don't know. It's hard to say.* | 4 | 2 |
| Absence of concerns | *I don't have any concerns. We just have to get on with it.* | 15 | 9 |

Associations between parents' and carers' socio-demographic characteristics and Covid-19-related changes, difficulties, and concerns, were explored using binary logistic regression analyses. Specific categories were selected for analysis based on the frequency of responses, and on the related potential effects on families' socio-emotional well-being in the short and long-term. Categories included in the analyses as outcome variables were: Family social isolation, Parents' emotional and psychological well-being, Children's emotional well-being and development, and Positive changes in family life. Although we recognised that Decreased stability of material resources could also be important for families' socio-emotional well-being, we did not select this category, as the demographic characteristics considered as predictors included employment status and having or not medical card (which has been means tested, providing insight on the families' financial situation).

**Table 4. Logistic regression analyses for demographics predicting changes related to the Covid-19 pandemic.**

| | Social isolation | | Children's emotional well-being and development | | Parents' emotional and psychological well-being | | Positive changes | |
|---|---|---|---|---|---|---|---|---|
| | B (*SE*) | OR (95% CI) | B (*SE*) | OR (95% CI) | B (*SE*) | OR (95% CI) | B (*SE*) | OR (95% CI) |
| Woman | 1.36 (1.16) | 3.91 (0.41–37.66) | [a] | [a] | [a] | [a] | 0.62 (1.12) | 1.87 (0.21–16.8) |
| Age | 0.05 (0.04) | 1.05 (0.98–1.13) | -0.04 (0.07) | 0.96 (0.84–1.09) | 0.07† (0.03) | 1.07 (1.00–1.14) | 0.08* (0.04) | 1.08 (1.00–1.17) |
| Born in Ireland | 0.31 (0.58) | 1.37 (0.44–4.23) | -0.12 (0.88) | 0.89 (0.16–5.04) | 0.82 (0.56) | 2.26 (0.75–6.80) | 0.31 (0.61) | 1.36 (0.41–4.55) |
| ISCED 6–8 | 0.13 (0.44) | 1.13 (0.48–2.67) | 1.11 (0.76) | 3.02 (0.69–13.27) | 0.60 (0.40) | 1.82 (0.82–4.02) | -0.2 (0.46) | 0.82 (0.33–2.04) |
| In paid employment | -1.29** (0.45) | 0.28 (0.11–0.66) | -1.41† (0.75) | 0.24 (0.06–1.05) | -0.27 (0.41) | 0.76 (0.34–1.69) | 0.49 (0.48) | 1.63 (0.64–4.14) |
| No medical card | 1.05* (0.47) | 2.87 (1.14–7.18) | 1.17 (0.82) | 3.21 (0.64–16.07) | 0.03 (0.41) | 1.03 (0.47–2.29) | -0.17 (0.46) | 0.84 (0.34–2.07) |
| Number of children | 0.04 (0.20) | 1.05 (0.70–1.55) | 0.08 (0.36) | 1.08 (0.53–2.21) | -0.13 (0.20) | 0.88 (0.59–1.30) | -0.06 (0.23) | 0.94 (0.6–1.46) |
| Number of adults | -0.33 (0.30) | 0.72 (0.40–1.30) | -0.18 (0.47) | 0.84 (0.33–2.12) | -0.03 (0.25) | 0.97 (0.60–1.58) | -0.11 (0.29) | 0.89 (0.5–1.59) |
| Constant | -3.53 (1.88) | 0.03 | -1.3 (2.66) | 0.27 | -3.53 (1.47) | 0.03 | -4.78 (1.98) | 0.01 |
| N | 147 | | 147 | | 147 | | 147 | |
| $\chi^2$ (df) | 16.264* (8) | | 7.775 (7) | | 9.945 (7) | | 6.286 (8) | |
| -2LL | 155.827 | | 65.286 | | 172.753 | | 145.161 | |
| Nagelkerke R$^2$ | .152 | | .132 | | .092 | | .065 | |

[a] Predictor variable not included given that only women scored 1 in the outcome variable.

†p < .10

*p < .05

**p < .01

The demographic characteristics considered as predictors were the following: participant's gender (woman, or man), age, country of birth (Ireland, or other), highest level of education completed (ISCED 6–8, or ISCED 3–5 or below), and work situation (in paid employment, or not in paid employment); existence of medical card in the family (no medical card, or having GP/full medical card); number of children in the participant's household; and number of adults in the participant's household. The highest level of education completed, work situation, and the existence of medical card were recoded into dichotomous variables given the small number of responses in some of the answer options. The receipt of social welfare payments by the family was not included as a predictor in the analyses, because the types of payments referred to by the participants were diverse, and did not suggest a common experience amongst the participants.

The assumptions of absence of multicollinearity, and linearity of the logit were met for the logistic regression analysis. Results indicated that some socio-demographic characteristics were found significantly or partially significantly associated with some of the selected categories.

The results regarding the associations between socio-demographic characteristics and the changes experienced during the pandemic are shown in Table 4. Participants who were in paid employment had a significantly lower probability of reporting Social isolation, and a partially significantly lower probability of reporting Negative changes in their children's emotional well-being and development, compared to participants that were not in paid employment. Participants who did not have a medical card had a significantly higher probability of reporting Social isolation, compared to those that had a medical card. The probability of reporting Negative changes in parents' emotional and psychological well-being was found to increase with age, although this association was only partially significant. The probability of reporting Positive changes for family or personal life was found to significantly increase with age.

**Table 5. Logistic regression analyses for demographics predicting difficulties related to the Covid-19 pandemic restrictions.**

| | Social isolation | | Childrens' emotional well-being and development | | Parents' emotional and psychological well-being | |
|---|---|---|---|---|---|---|
| | B (SE) | OR (95% CI) | B (SE) | OR (95% CI) | B (SE) | OR (95% CI) |
| Woman | 1.16 (0.78) | 3.19 (0.7–14.61) | 0.03 (1.15) | 1.03 (0.11–9.86) | a | a |
| Age | 0.04 (0.03) | 1.04 (0.97–1.10) | -0.04 (0.04) | 0.96 (0.88–1.05) | 0.06 (0.04) | 1.06 (0.97–1.15) |
| Born in Ireland | -0.18 (0.48) | 0.83 (0.33–2.12) | 1.75 (1.07) | 5.73 (0.71–46.29) | 0.77 (0.68) | 2.17 (0.57–8.21) |
| ISCED 6–8 | -0.33 (0.39) | 0.72 (0.34–1.53) | 0.39 (0.52) | 1.48 (0.53–4.13) | 0.23 (0.48) | 1.26 (0.50–3.21) |
| In paid employment | 0.09 (0.37) | 1.09 (0.53–2.28) | -0.10 (0.5) | 0.90 (0.33–2.42) | 0.74 (0.51) | 2.10 (0.77–5.70) |
| No medical card | 0.38 (0.38) | 1.47 (0.70–3.09) | 0.25 (0.52) | 1.28 (0.46–3.56) | 0.17 (0.47) | 1.19 (0.47–3.02) |
| Number of children | -0.13 (0.18) | 0.87 (0.62–1.23) | 0.08 (0.24) | 1.09 (0.68–1.72) | -0.16 (0.26) | 0.85 (0.51–1.41) |
| Number of adults | 0.07 (0.22) | 1.07 (0.70–1.64) | -0.23 (0.31) | 0.80 (0.43–1.47) | -0.21 (0.31) | 0.81 (0.44–1.49) |
| Constant | -1.84 (1.48) | 0.16 | -1.84 (2.19) | 0.16 | -4.02 (1.75) | 0.02 |
| N | 150 | | 150 | | 150 | |
| $\chi^2$ (df) | 5.798 (8) | | 5.724 (8) | | 8.663 (7) | |
| -2LL | 197.616 | | 122.810 | | 135.743 | |
| Nagelkerke $R^2$ | .051 | | .065 | | .091 | |

[a] Predictor variable not included given that only women scored 1 in the outcome variable.

The results regarding the associations between socio-demographic characteristics and the difficulties experienced during the pandemic are shown in Table 5. No significant or partially significant associations were found.

The results regarding the associations between socio-demographic characteristics and concerns related to the pandemic are shown in Table 6. Participants who were in paid employment had a significantly lower probability of reporting Social isolation, compared to those that were not in paid employment. The probability of reporting Social isolation was found to significantly decrease when the number of adults living in the household increased. Participants

**Table 6. Logistic regression analyses for demographics predicting concerns related to the Covid-19 pandemic.**

| | Social isolation | | Children's emotional well-being and development | |
|---|---|---|---|---|
| | B (SE) | OR (95% CI) | B (SE) | OR (95% CI) |
| Woman | 0.05 (0.94) | 1.05 (0.17–6.60) | 0.82 (1.12) | 2.28 (0.25–20.51) |
| Age | -0.01 (0.04) | 0.99 (0.91–1.07) | -0.02 (0.04) | 0.98 (0.9–1.07) |
| Born in Ireland | -0.88 (0.53) | 0.42 (0.15–1.19) | -0.03 (0.60) | 0.97 (0.3–3.13) |
| ISCED 6–8 | 0.46 (0.48) | 1.58 (0.61–4.07) | 0.13 (0.50) | 1.14 (0.43–3.03) |
| In paid employment | -0.93* (0.47) | 0.39 (0.16–0.98) | 0.45 (0.54) | 1.57 (0.55–4.50) |
| No medical card | 0.11 (0.48) | 1.11 (0.43–2.86) | 1.19* (0.54) | 3.28 (1.13–9.50) |
| Number of children | -0.39 (0.25) | 0.68 (0.41–1.11) | -0.11 (0.28) | 0.90 (0.52–1.55) |
| Number of adults | -0.69* (0.34) | 0.50 (0.25–0.98) | 0.03 (0.31) | 1.03 (0.57–1.88) |
| Constant | 2.05 (1.78) | 7.8 | -2.51 (2.02) | 0.08 |
| N | 146 | | 146 | |
| $\chi^2$ (df) | 13.640† (8) | | 9.674 (8) | |
| -2LL | 137.332 | | 124.014 | |
| Nagelkerke $R^2$ | .138 | | .107 | |

†$p < .10$

*$p < .05$.

who did not have a medical card had a significantly higher probability of reporting concerns with Childrens' emotional well-being and development, compared to those that had a medical card (General Practitioner only or full coverage).

## Discussion of results

Study 1 aimed to explore the views of parents and carers of young children (up to six years old) about changes experienced with the Covid-19 pandemic, difficulties regarding the restrictions, and current and future concerns about Covid-19. Furthermore, this study aimed to explore what socio-demographic characteristics were associated with changes, difficulties, and concerns related to family well-being.

Social isolation, Negative impact on parent's emotional and psychological well-being, and Children's emotional well-being and development were identified as families' experiences during the pandemic. These findings were aligned with previous studies that described negative impacts on parents' and children's socio-emotional well-being [6, 11]. Efforts to promote mental health can positively impact parents', carers' and children's well-being [24]. Given the likely decrease in social interactions and social networks due to the pandemic, measures to promote social connections and prevent mental health issues, including loneliness, from becoming chronic can be needed [25]. According to our findings, these measures can be particularly relevant for families whose parents are not in paid employment.

In the current study, a negative impact on parents' emotional and psychological well-being was only reported by women. We were not able to explore if gender significantly predicted these experiences via the regression analyses. The study sample had an uneven gender distribution, and was mainly constituted by women. Future research on parenting and caregiving should endeavour to include fathers, since mothers have made up most of the participants in previous studies [26].

The category Negative impact on parent's emotional and psychological well-being included increased demands on parents in terms of childcare and housework responsibilities. These were found to increase during the pandemic, particularly for women [27, 28]. Irish research prior to the pandemic indicated differences in the time dedicated to childcare between men and women. According to the European Quality of Life Survey from 2003 to 2016, of the population providing regular childcare (at least once a week) in the Republic of Ireland, the mean weekly time was 42.6 hours for women and 25.2 hours for men [29]. With the continuation of remote working, policies addressing childcare and gender equality can play a major role in promoting parents' mental health and families' well-being.

Conversely, 20% of the parents and carers in this study affirmed experiencing positive changes during the pandemic, including with working from home and spending more time with children. Positive changes in family life were also reported in more studies. In the Republic of Ireland, some parents of children aged between one and 10 years reported that positive aspects of lockdown included increased opportunities for children to play with siblings, alone, and outdoors, and a break from the usual routine [11]. A German study found that parents, including those of preschool children, indicated positive aspects such as a slower pace of life and increased family time [8]. In Italy, parents reported rediscovering the value of being together, and improving cohesion, expressiveness and positive parenting [30]. Despite all the negative effects caused by the Covid-19 pandemic, interventions should seek to sustain the positive changes for families. This may require policy changes, workforce development plans and technical support to enable remote work that sufficiently recognise the hybrid working context.

Structural restrictions were the changes cited most frequently, and current and future concerns included a focus on Physical health, and Uncertainty about the future. The findings

obtained need to be framed in the period of the data collection, which was carried out at the beginning of the third wave of the pandemic in the Republic of Ireland, around eight months after the first restrictions were implemented, and before the start of the vaccination programme. Restrictions and uncertainty were also identified as causes of stress by parents some months after the pandemic started in Germany [8]. Given the possibility of new variants of the virus and changes in the infection rates, governments and health professionals may need to consider investments in further support for families in coping with the risk of infection and related uncertainty.

The findings of this study highlighted the importance of considering socio-demographic characteristics of the parents and families when analysing the Covid-19-related experiences, such as parents' age, work situation, socio-economic situation, and family composition. Tailored measures to enable the maintenance of social connections can be particularly important to parents and carers not employed and single-parent families [9, 14]. According to the results obtained, families with parents or carers not in paid employment can also benefit from further support regarding the promotion of their children's emotional well-being and development.

Participants who did not have a medical card had a higher probability of reporting social isolation as a change, and concerns with children emotional wellbeing and development, compared to those that had a medical card, who could be in a more disadvantaged financial situation. This is despite the expectation that pandemic-related challenges could be more accentuated for households with lower income [14]. Previous research indicated that middle class parents were more likely than working class parents to receive online support from childcare providers during the pandemic [10]. Financial resources can impact on access to supports, and these became even more important during the pandemic [31]. However, these supports might not be sufficient to mitigate perceptions and experiences of social isolation, and concerns with children resulting from Covid-19.

Previous research suggested that pandemic-related negative effects on families may be attributable less to general socio-demographic or socio-economic factors, and more to pandemic-specific socio-economic factors, such as job losses and financial difficulties [8]. However, pandemic-related characteristics (including income) were not always found associated with an increase in parents' negative affect [32]. Within the current study, the questions applied to collect socio-demographic data did not allow us to identify if the work situation and existence of medical card changed or not due to the pandemic. More research on the effects of both general socio-demographic and pandemic-specific socio-economic factors on families can contribute to better understand their specific needs.

The probabilities of reporting both Negative changes in parents' emotional and psychological well-being, and Positive changes in family or personal life were found to increase with the parent's or carer's age. In a previous study, symptoms of depression during the pandemic increased with parents' age [5]. Further research exploring age-related factors that can affect parents' experiences during and after the pandemic, can contribute to understand how to promote parents' and families' well-being.

According to the findings of this study, policies and practices aimed at providing social-emotional support to families, including measures to support work-family life balance, can be relevant to improving family's well-being, and reinforcing the positive changes in family and personal life reported. Future research exploring the views and experiences of families in subsequent phases of the pandemic can be important to better understand its medium- and long-term effects.

In terms of limitations of the study, we are aware that the primary focus of the questionnaire was to evaluate a parenting support programme. Adding Covid-19 related questions meant that it took around 20 minutes to complete, which may have influenced participants'

responses. We acknowledge that other methods could have been applied to grouping the codes into categories. For instance, social isolation could have been incorporated into emotional well-being, but we chose to analyse these as separate categories, as less interaction with others does not necessarily imply negative psychological or emotional impacts. We opted to use conceptual content analysis, considering that participants' responses were generally short (one or two sentences in most cases). Future research might consider other methods, such as relational content analysis and thematic analysis, to understand the relationship between the issues expressed.

## Study 2

The objective of Study 2 was to explore the views and experiences of children (eight-17 years old), parents with children aged under 18 years, and service providers working with families about the impact of the pandemic on families' life, and relevant supports.

### Methodology

**Participants.** Participants in the study were children, parents, and service providers in Dublin 24, a designated disadvantaged area according to the Pobal HP Deprivation Index. Only children and parents living in the area were included as participants while service providers had to be directly working with children and families in the defined community to be included. Fifty children (27 females and 23 males) participated in six Focus Group Discussions (FGDs). The first set of three FGDs included 27 children from eight to 11 years old. The second set of three FGDs included 23 children from 14 to 17 years old.

Thirty-five children lived with both parents while 15 children lived with one parent. Twenty-five children had both parents employed, 21 children had one parent employed and four children had no parents employed. For ethical reasons, demographics were not collected for parents and service providers. Seventeen parents participated in two FGDs, and 20 service providers participated in two FGDs. Service providers worked in a range of sectors, including family support, speech and language therapy services, community work, migrant and refugee services, education, primary care psychology, child protection, drug and substance abuse services and youth work. These were deduced from the service providers introductions at the FGDs.

**Instruments.** Using FGD guides, data on the experiences and impact of Covid-19 were collected using the following open-ended questions for all the three groups of participants (children, parents and service providers): How has the Covid-19 pandemic affected the lives of children and families (mentally, social life, education, and economically /financially)? What support services should be provided to children and families affected by the Covid-19 pandemic?

**Data collection.** Participants were recruited through gatekeepers, drawn from local services. Gatekeepers for children and parents included schools, child and family support services, youth organisations, and Early Learning and Care services. Service providers were recruited through the Childhood Development Initiative (CDI) contact database and through the South Dublin Children and Young People's Service Committee (CYPSC). Data were collected via FGDs, between October 2020 and December 2020. In the Republic of Ireland, this period corresponded to the end of the second wave and the beginning of the third wave of the Covid-19 pandemic [22]. Primarily, the study sought to explore the experiences of child poverty in Dublin 24 and its impact on children and families. However, to respond to the emerging pandemic, the study extended its focus to explore the impact of Covid-19 on children and families.

**Ethical considerations.** Written ethics approval for the study was obtained from the national Child and Family Agency's (Tusla) Research and Ethics Committee. Participants

were provided with age-appropriate information sheets, consent, and assent forms written in simple English. The information on the study provided included the reasons for the research, data treatment procedures, and researcher's role and contact. The research team sought to ensure that all participants understood that they could refuse to participate or leave the study at any time, with no consequences. Before data collection, written informed consent was obtained from participating adults, assent from participating children, and written consent from children's legal guardians. Infection prevention practices related to Covid-19 were considered during data collection, following the current public health measures. During data analysis, all data were anonymised by removing all potential personal identifiers.

**Reflexivity.**   The author who conducted the FGDs and was involved in coding the data (JS) was a male Data Specialist with a Masters in Applied Social Research, and with previous training in analysing qualitative data. During data collection and analysis, the author endeavoured to avoid imposing personal assumptions about families' and poverty-related experiences. The consolidated criteria for reporting qualitative research checklist (COREQ) guided the reporting of this study [33].

**Data analysis.**   With participants' consent and assent, FGDs were audio recorded. All audios were transcribed verbatim, checked for accuracy, and coded manually for analysis. Data analysis followed the six steps of thematic analysis [34]. After transcribing audio files, all transcripts were read in detail multiple times by one of the authors, to fully understand the data. After reading the transcripts, data were coded and later organised into categories pulling together the similar categories. Data categories were read again and organised into emerging themes. Themes were revised and refined into final themes emerging from the data. Although the analysis focused on the recurring themes in the data, particular attention was given to examining negative or rare contributions during the FGDs.

Demographic data for participants were entered into an Excel file for management and analysis.

## Results

Three main themes were identified, namely: education during the pandemic; unmasking food poverty; and children's socio-emotional health. Within the education during the pandemic theme, two sub-themes were identified: challenges with online education was identified by parents and service providers, while the sub-theme on uncertainty of children's education was named by children and service providers. The theme on unmasking food poverty was identified by service providers and the theme on children's socio-emotional health was articulated by all three participating groups.

**Education during the pandemic.**   Participants shared their experiences of education during the pandemic, pointing to challenges with access to online education and the uncertainty which the Covid-19 restrictions had brought to children's education.

*a. Challenges with online education.* With the introduction of Covid-19 restrictions in Ireland, schools closed and moved online. Not all children, however, could afford to access online education with major challenges being experienced in relation to affordability of devices and access to the internet at home. Parents expressed frustration with the extra costs of purchasing devices for children's online education. Some parents shared the same device for their own work or education, and their children's education.

> *"I have had to put extra pressure on myself now for Christmas because I have got to get them tablets obviously to do their schoolwork online. That is an extra 180 to 200 euro each that we genuinely do not have."*—**Parent**

*"Most of the time I do not have Wi-Fi. I put the hotspot on my phone and that is just what we must do. . .but it is just sad to think that there are so many kids out there that do not have access to the things that they need for their education."*—**Parent**

Service providers echoed these issues with some providing laptops to families for children to access online education.

*"The older kids were doing school online and an awful lot of them did not have laptops. . .. So, we would lend some people laptops for the whole period of lockdown, to try and at least solve that side of things."*- **Service Provider**

The closure of services and shift to online education also affected some parents, particularly those who were pursuing further educational programmes to enhance the socio-economic status of their families. These parents experienced similar challenges to those affecting children (access to internet and devices), with the resultant impact on parents' ability to pursue educational programmes and re-enter the employment sector.

*"We have a number of young women that we are helping. . ..—that are doing the programme. Some of the programmes are online, but we're finding that people don't have access to laptops. They're hoping to engage in education and go back into training, or employment opportunities but Wi-Fi is a barrier."*—**Service Provider**

*"We have with three young women that we are delighted to get to support in their education, unless we get laptops for them now, they are not going to be able to do it. So, it is an issue for people in the area."*—**Service Provider**

*b. Uncertainty of children's education.* With the closure of schools, some children expressed uncertainty about the resumption of normal education, with some children having missed several days of school. Children felt they had been left behind due to the amount of school days they had missed because of Covid-19 and were worried about how they would catch up with their classwork.

*"It has impacted my life because I missed a year of school. You cannot go back from that; you cannot go back for like for classes. So, it impacted my school."*—**Child**

*"Because of school. We have missed days of school. You cannot go to school."*—**Child**

From the perspective of service providers, the closure of schools removed an important support system for children at risk of dropping out. With the closure of schools, many service providers were worried that some children would not return to education, with long term impacts on their future.

*"The children are fearful that the school might close again, and they must manoeuvre school in a very different way. That feeling for children of being put into uncertain position and not knowing what the outcomes might be."*—**Service Provider**

*"Some children that were just hanging on in their secondary school. And then Covid struck. . ...(they) are lost now and what's out there for them now . . .school was kind of the right place for them. They will not get those children back to school, so that teenage group is very vulnerable as well."*—**Service Provider**

**Unmasking food poverty.** For many service providers, food poverty was *"hidden"* before the pandemic and the depth of it was *"unmasked"* following the outbreak of Covid-19 and subsequent restrictions on services, particularly school closures. The extent and depth of how families experienced food poverty since the outbreak of the pandemic was a surprise for many service providers.

> *"While I would be aware of the food poverty in the area I think what really struck me was when we made up the resource packs to bring to children,. . . the awareness also of the real lack of resources in the home for them."*—**Service Provider**.

Before the pandemic, children accessed at least one nutritious meal in school or EL C service, and services were aware that some children brought food home for their siblings. This had "*masked*" the depth of food poverty experienced by families. However, this service was interrupted during the Covid-19 pandemic resulting in an increase in food poverty for many families. This was further intensified by Covid-19 related job losses and struggles to access adequate incomes.

> *"We would often get lunches from the school, whatever's leftover, and they'd be shoving them into their bags to bring home for later on or for their siblings."*—**Service Provider**

> *"We would all have had the school meals for children (breakfast clubs and lunch at school). . .. and after school and they would have gotten an after-school snack. So, because children were at home all the time and they were obviously eating more and that support was taken away."*—**Service Provider**

With the cessation of school meal supports for children and awareness that food poverty spiked in many families, service providers immediately responded, in a multi-agency and coordinated way, by providing food packs and deliveries for those families most impacted by food poverty.

> *"Obviously the most vulnerable families and the highest need ones were the ones that we met first, but it was all the multi-agency response from the [Community Centre]. And I am not saying no family went without food, but I will say every family that came to our attention got food and got all hot meals and got packs. . ."*—**Service Provider**

**Children's socio-emotional health.** Covid-19 restrictions had a negative impact on children's socio-emotional well-being, due to the limitations on their outdoor activities and closure of most services. Children missed spending time with their friends. Although schools opened in September 2020, having closed in March 2020, schools had to implement new guidelines including the formation of 'bubbles' or 'pods' with no or limited mixing outside of those groups, and changes to the physical environment. This limited childrens' freedom to mix and play with each other.

> *"I cannot play with him or him anymore because we are in two different classes now, we got split up and we cannot even play with each other."*—**Child**

> *"Yeah, because young people used to be able to go outside like spend time with their friends and do sports. But until then all got closed and it was not safe for them to go out and visit your friends."*—**Child**

With limited outdoor activities and reduced interactions with their friends, children expressed concerns that the pandemic might affect them mentally and socially. They expressed concerns about becoming depressed due to the pandemic, and fear of passing on the coronavirus to their family members as schools re-opened.

*"Yes, maybe depressed. Well, I cannot do anything, I must remain inside. I cannot even do anything with my friends, I cannot go to school. One of your family members may get the coronavirus and they got sick."*—**Child**

Parents articulated concerns that the Covid-19 pandemic and subsequent restrictions had impacted on children's social development and children's relationships with their peers. Some parents were concerned about the apparent regression in their children's social development.

*"Right now, it is the lockdown because my kids do not go out to make new friends. They depend a lot on other kids for reassurance; he is afraid to go to a new setting where it is not the people that he is friends with, like he is not confident in making relationships."*—**Parent**

Service providers reported an increase in general anxiety, speech and behavioural difficulties linked to the Covid-19 pandemic, especially with the re-opening of schools. As noted by children, service providers also reported that children living in intergenerational households were *'fearful'* of passing on Covid-19 to their grandparents, and some children were found to manifest a range of behaviours as a result of increased anxieties.

*"Some of our children have moved back in with grandparents. . ...or their aunties and uncles are still living in the homes as well. So, it was huge anxiety we found with the kids, and they are still talking about passing Covid on to their grandparents."*—**Service Provider**

*"I took a drop-in call, and it was a child presenting with a stutter or stammer that, the mom would say, is pandemic-related and particularly returning to school."*—**Service Provider**

Some service providers observed that Covid-19 had contributed to reports of self-harm among children.

*"My experience in the last six weeks, the parents and grandparents that are getting in contact, very concerned about their teenage daughters who are self-harming, who never self-harmed before."*—**Service Provider**

## Discussion of results

This study explored how the Covid-19 pandemic affected children and families in Dublin 24, a designated disadvantaged area on the Pobal HP Deprivation Index [19]. Overall, the study found Covid-19 to be more than a physical health issue as it affected the material, educational, socio-emotional and mental well-being of children and families. Across Europe and globally, the Covid-19 pandemic widened educational inequalities in communities [31, 35]. Access to digital technology for online education has been noted as a key challenge. The Growing Up in Ireland survey for 2021 [31] found that less than 20% of 12-year-old children always had access to online education. More children are at risk of missing out on education as the pandemic continues to present challenges globally [31]. Across Europe, the use of digital technologies in healthcare and education is likely to increase in the future with the acceleration of the digital transformation and investments which were made during the pandemic [35]. Studies from

other countries showed that online education during the pandemic negatively affected children's performance and that many children were dissatisfied with online education [36]. New waves of the pandemic have swept across Europe and globally, and children have missed out on school time because of Covid 19 illness or being close contacts in the family. Systematic measures should be implemented to ensure children have access to online education (when they cannot attend classroom education), and support systems are required for children who have missed school days, to ensure they do not fall behind their peers.

Food poverty was identified as a major challenge for some families, and this was exacerbated by the Covid-19 pandemic. Through the Roadmap to Social Inclusion, the Government of the Republic of Ireland commits to providing income supports for families and expanding the School Meals Programme [37]. The European Child Guarantee encourages EU Member States to support access to healthy meals during school or early years services, including through in-kind or financial support [38], although these recommendations have not yet commenced implementation. While most schools and Early Learning and Care services have provided children with at least one nutritional meal a day, this support was removed when schools closed. Therefore, support and systems need to be developed and implemented to ensure that children have a nutritious meal every day, in and out of school.

In this study, data were gathered from several groups of participants (children, young people, parents, and service providers), with ten FGDs being conducted in total, allowing different experiences and perspectives to be explored. FGDs for children were undertaken with different age groups, allowing for appropriate questions to be asked of each cohort, tailoring the methodology and enhancing the quality of the discussions. The inclusion of children's perspectives in research and policy development is recognised as being crucial to ensuring relevant responses to their specific needs [39], and yet this is still not common practice [40].

One of the methodological limitations of the study relates to Covid-19, with some FGDs not implemented due to health restrictions. Some of the techniques for children's participation (art and games) could not be implemented due to the risk of transmission of Covid-19.

## General discussion

The overall objective of both studies was to understand the views and experiences of families during the Covid-19 pandemic, particularly focusing on families living in areas characterised by socio-economic disadvantage. Study 1 focused on the changes, difficulties, and concerns of parents and carers of children up to six years old, and on exploring associations with their socio-demographic characteristics. Study 2 focused on the views and experiences of children, from eight to 17 years old, parents, and service providers working with families.

Both studies identified the impact of the pandemic on families' social isolation, and socio-emotional well-being. In Ireland, studies have observed the socio-emotional challenges experienced by children during the pandemic, including low mood, depressive symptoms, social isolation, anxiety, and increases in maladaptive behaviour [31, 41]. Restrictions and lockdowns were found to increase anxiety and loneliness among adolescents, and negative behaviour in younger children [41]. Previous research indicated that parents of young children experienced loneliness during the pandemic [42], and were concerned about the social and emotional development of their young children, particularly when unable to attend childcare services [10, 11].

Parents' views and experiences during the pandemic can affect not only their mental health, but also their children's outcomes. Parents' stress was reported to negatively affect children's behaviour and emotional management [5, 6]. Additionally, lower-quality parenting during the

pandemic was found to be predicted by caregiver depression, higher number of children in the home, unmet childcare needs, and relationship distress [24].

The findings described highlighted the importance of regular assessment of both short- and long-term impacts of Covid-19 on children's, parents' and carers' mental health and well-being, as well as the provision of relevant support. Specifically, evidence-informed services to address children's and parents' anxiety and stress can be relevant to integrate across a range of community services, together with support for frontline professionals delivering these services. The experiences regarding social isolation also reinforced the importance of ensuring that families have the resources to maintain social connectedness, such as digital access, while allowing for physical distance when needed.

Social supports during the pandemic were found positively associated with families' socio-emotional well-being. In a previous study with parents and adolescents, the two most important protective factors regarding psychological stress were social support and keeping busy during lockdown, while the most significant risk factors were loss of mobility and social isolation [43]. Strains on parent capacities were also found to include limited social supports, as well as psychological distress and too much unstructured time [28].

In regard to childcare services and schools, our findings reinforce that these settings provide an important nurturing environment which supports children's socio-emotional development [11]. If different educational outcomes for children based on socio-economic background are to be avoided, targeted measures are required which directly respond to the gaps in access and provision experienced by poorer children. Since families in disadvantaged contexts are less likely to have adequate broadband and hardware [31], often used for remote learning and contact with Early Learning and Care services and schools, these digital tools need to be accessible to all families. Learning supports based on an early assessment of children's achievements and difficulties, adequate training and resources for teachers, and promotion of parent engagement should also be considered [12, 44].

Uncertainty about the future was identified by young people in Study 2 as a response to educational restrictions, resulting in concern about their capacity to complete their education. These concerns were mirrored by parents in Study 1, who also expressed uncertainty about both their own future and that of their children. Adequate resources (such as digital tools) should be made available to create equal access to education for young people. Parents and young people should also be supported to access information and provided with opportunities to name their worries, and develop coping mechansims to manage anxiety. The implemented measures should particularly focus on the groups less likely to return to school, particularly girls, children with disabilities, and those in poorer or marginalised families [45]. Future research could assess the extent to which concerns regarding uncertainty have been addressed by the lifting of the restrictions.

Both studies indicated that socio-demographic characteristics of the families could influence their pandemic-related experiences. Study 1 showed associations between Covid-19-related experiences and parents' age, work situation, socio-economic situation (as suggested by the existence or not of medical card), and family composition regarding the number of adults in the household. In particular, participants who were not in paid employment had a higher probability of reporting social isolation, and negative changes in their children's emotional well-being and development, compared to those who were in paid employment. Study 2 highlighted that inequity in relation to children's education and access to a nutritional diet was exacerbated during the pandemic related restrictions. Inability to meet financial obligations and maintain social ties were found to be significantly associated with increased reporting of family stress [46]. In a previous study, sex, age and parent's employment status were found to have a predictive value on parents' internalising symptoms, such as stress and anxiety [5].

Consideration of socio-economic characteristics is vital to enabling effective responses and combatting the increase in inequalities. Mechanisms at play in each culture and country, as well as social inequalities and living conditions need to be considered when developing lock-down measures and mitigation policies [14, 47].

The studies discussed here reinforced the importance of implementing measures to support families, both in terms of targeted policies to preserve or increase economic security and stability of employment, as well as to promote parents' and children's socio-emotional well-being, and children's development, and combat families' social isolation. In this sense, a systematic approach needs to be considered to address the negative impacts of Covid-19, requiring coordination of interventions across a range of sectors [48].

Findings from Study 1 also indicated that some parents and carers experienced positive changes in their personal or family life, particularly having more time to spend with their family. Research focusing on both negative and positive responses of children and families to the pandemic can contribute to understanding the impact of modern daily life and environments on mental health and well-being [49]. For example, how we can help parents to improve the balance between their work, and personal and family life, sustaining the positive changes regarding the time spent with family achieved during the pandemic; and for those families who have not experienced this positive change, what it will take to enable them to spend more time with their children and family.

Research on both the short- and long-term effects of the pandemic has the potential to increase our understanding of families' needs and resources. Identifying children, young people and parents' views and experiences can inform policy and practice aimed at supporting families, namely the types of interventions needed to address emerging needs or existing inequalities which were exacerbated and unmasked by the pandemic.

## Acknowledgments

We wish to acknowledge our gratitude to all children, young people, parents, carers, and service providers who participated in this research. It would not have been possible without your participation. To our "gatekeepers", namely schools, local organisations, and early learning and care services, including the Parent/Carer Facilitators, who helped us access and recruit participants, we will always be thankful. We also thank all the colleagues from the Childhood Development Initiative for the continuous support throughout the research.

## Author Contributions

**Conceptualization:** Catarina Leitão, Jefrey Shumba, Marian Quinn.

**Data curation:** Catarina Leitão.

**Formal analysis:** Catarina Leitão, Jefrey Shumba.

**Funding acquisition:** Catarina Leitão, Jefrey Shumba, Marian Quinn.

**Investigation:** Catarina Leitão, Jefrey Shumba.

**Methodology:** Catarina Leitão, Jefrey Shumba.

**Project administration:** Catarina Leitão, Jefrey Shumba, Marian Quinn.

**Resources:** Catarina Leitão, Jefrey Shumba, Marian Quinn.

**Supervision:** Marian Quinn.

**Validation:** Jefrey Shumba, Marian Quinn.

**Writing – original draft:** Catarina Leitão, Jefrey Shumba.

**Writing – review & editing:** Marian Quinn.

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
