## [Decision Letter · Decision Letter 0]

14 Apr 2022

PONE-D-21-40789Perspectives and experiences of Covid-19: Two Irish studies of families in disadvantaged communitiesPLOS ONE

Dear Dr. Leitao,

Thank you for submitting your manuscript to PLOS ONE. After careful consideration, we feel that it has merit but does not fully meet PLOS ONE’s publication criteria as it currently stands. Therefore, we invite you to submit a revised version of the manuscript that addresses the points raised during the revision process.

 You should especially note the points about clarification and elaboration of your methods and stratification.

Please submit your revised manuscript by May 29 2022 11:59PM.  If you will need more time than this to complete your revisions, please reply to this message or contact the journal office at plosone@plos.org. Please include the following items when submitting your revised manuscript:A rebuttal letter that responds to each point raised by the academic editor and reviewer(s). You should upload this letter as a separate file labeled 'Response to Reviewers'.A marked-up copy of your manuscript that highlights changes made to the original version. You should upload this as a separate file labeled 'Revised Manuscript with Track Changes'.An unmarked version of your revised paper without tracked changes. You should upload this as a separate file labeled 'Manuscript'.If applicable, we recommend that you deposit your laboratory protocols in protocols.io to enhance the reproducibility of your results. Protocols.io assigns your protocol its own identifier (DOI) so that it can be cited independently in the future. For instructions see: https://journals.plos.org/plosone/s/submission-guidelines#loc-laboratory-protocols. Additionally, PLOS ONE offers an option for publishing peer-reviewed Lab Protocol articles, which describe protocols hosted on protocols.io. Read more information on sharing protocols at https://plos.org/protocols?utm_medium=editorial-email&utm_source=authorletters&utm_campaign=protocols.

We look forward to receiving your revised manuscript.

Kind regards,

Gerard Hutchinson, MD

Academic Editor

PLOS ONE

Journal Requirements:

The authors have read the journal’s policy and have the following competing interests: Study 1 was conducted within a project that has received funding from the European Union’s Horizon 2020 research and innovation programme under the Marie Skłodowska-Curie grant agreement No 890925. Study 2 was funded by Tusla under the Area Based Childhood funding and the Child and Youth Participation Initiatives grant.

Reviewers' comments:

Reviewer's Responses to Questions

**Comments to the Author**

1. Is the manuscript technically sound, and do the data support the conclusions?

Reviewer #1: Yes

2. Has the statistical analysis been performed appropriately and rigorously? 

Reviewer #1: Yes

3. Have the authors made all data underlying the findings in their manuscript fully available?

Reviewer #1: Yes

4. Is the manuscript presented in an intelligible fashion and written in standard English?

Reviewer #1: Yes

5. Review Comments to the Author

Reviewer #1: Thank you for the opportunity to review this interesting paper. The paper presents two studies investigating disadvantaged families in the Republic of Ireland. To provide context for my comments I will make suggestions under headings.

Introduction

You note that restrictions on public life can have a negative impact on individual and family lives. Can you specify some of these impacts because you then go on to note research about the start of the pandemic but I was unsure if the data presented in that section relates to the mandated closures (lockdown) or the pandemic more broadly. (looking for more specificity in your statements through this section).

I'm curious about the decision to add both studies to your submission. The studies relate in topic but are not clearly consecutive or mixed methods where one informs the other. Either they have but it is not clear or you need to justify the decision in your paper.

Study 1

Uneven gender distribution that should be reflected in your discussion and comments.

I would consider adding reflexivity statements to both of your studies rather than just a statement regarding experience in the methodology.

line 306 you allude to positive change seen in other research but don't specify or consider how/why? How does your specific group compare?

The discussion largely repeated the results with brief comments made regarding the implications. You missed an opportunity to explore these alignment with research and recommendations specific to each study (though I note you include a larger discussion later). My advice is reduce replication of results section and extend implications and considerations in study 1 discussion.

Study 2

This is a difficult group to sample and you are commended for your efforts. Clarity earlier in the paper about the differentiation of focus groups by age might be useful as it is covered in the discussion but definitely made my list of questions early.

Results

All there groups appear to be presented together. This needs to be discussed and if sub themes are not addressed by all groups this identified. For example, it seems aspects of educational challenges due to device access were not addressed by children (or you don't have examples of them). Given you have decided to reflect all stakeholders together I suggest you qualify the groups that spoke to each theme and sub theme so we can better appreciate the views of all stakeholders as unique groups not just as a collective.

Some of the quotes are overly long and could be synthesised but still be representative. Also in some sections the quotes do not directly address the comments above it.

648 I suggest you draw specific attention to technology and WIFI access as clear examples of the divide you speak of

653 information as you suggest may not be enough without concrete resourcing. I would also note that international studies suggest that thee vulnerable groups are less likely to return to school so some of the 'impact' data you speak of is already there to reference.

Overall an interesting paper that really just needs elaboration, specificity and qualification in some areas.

A pleasure to read.

6. PLOS authors have the option to publish the peer review history of their article (what does this mean?). If published, this will include your full peer review and any attached files.

Reviewer #1: **Yes: **Jade Sheen

---

## [Author Response · Author response to Decision Letter 0]

27 May 2022

Responses to Academic Editor’s comments on Journal Requirements

1. Please ensure that your manuscript meets PLOS ONE’s style requirements, including those for file naming. The PLOS ONE style templates can be found at 

Response: We have checked the manuscript and filenames to ensure that they comply with PLOS ONE's style and naming requirements. We removed the italics style from the text presented in the Instruments sections. We added visible margins to all cells of the tables. Moreover, an additional private email address of the corresponding author CL on the title page has been added to ensure continuity in communication in cases of future inquiries about the manuscript. The reason is that the given institutional address will be deactivated in the near future.

We also revised the grammar of some sentences to improve clarity, and the use of capital letters and hyphenated words to ensure consistency throughout the manuscript.

The authors have read the journal’s policy and have the following competing interests: Study 1 was conducted within a project that has received funding from the European Union’s Horizon 2020 research and innovation programme under the Marie Skłodowska-Curie grant agreement No 890925. Study 2 was funded by Tusla under the Area Based Childhood funding and the Child and Youth Participation Initiatives grant.

Please confirm that this does not alter your adherence to all PLOS ONE policies on sharing data and materials, by including the following statement: “This does not alter our adherence to PLOS ONE policies on sharing data and materials.” (as detailed online in our guide for authors http://journals.plos.org/plosone/s/competing-interests). If there are restrictions on sharing of data and/or materials, please state these. Please note that we cannot proceed with consideration of your article until this information has been declared. 

Response: We updated the Competing Interests statement in our cover letter to confirm the adherence to all PLOS ONE policies on sharing data and materials. The complete Competing Interests statement is now the following: 

Study 1 was conducted within a project that has received funding from the European Union’s Horizon 2020 research and innovation programme under the Marie Skłodowska-Curie grant agreement No 890925. Study 2 was funded by Tusla under the Area Based Childhood funding and the Child and Youth Participation Initiatives grant. This does not alter our adherence to PLOS ONE policies on sharing data and materials.

Response: We will provide the relevant accession numbers or DOIs to access the data after acceptance, before publication. We do not wish to make changes to our Data Availability statement.

Response: We reviewed the reference list. We corrected and completed some references. We also removed the retracted references and replaced them with the relevant current references, so the revised manuscript does not include retracted articles. We also added new references as we edited the content following the Reviewer’s comments.

Responses to Reviewer’s comments to the Author

Reviewer #1: Thank you for the opportunity to review this interesting paper. The paper presents two studies investigating disadvantaged families in the Republic of Ireland. To provide context for my comments I will make suggestions under headings.

Introduction

You note that restrictions on public life can have a negative impact on individual and family lives. Can you specify some of these impacts because you then go on to note research about the start of the pandemic but I was unsure if the data presented in that section relates to the mandated closures (lockdown) or the pandemic more broadly. (looking for more specificity in your statements through this section).

Response: We added a paragraph (which corresponds to the second one of the introduction) on the impacts of the pandemic more broadly. In the following paragraph, we sought to provide more details on the impacts of the Covid-19 containment and mitigation measures found in the referenced studies.

Reviewer #1: I’m curious about the decision to add both studies to your submission. The studies relate in topic but are not clearly consecutive or mixed methods where one informs the other. Either they have but it is not clear or you need to justify the decision in your paper.

Response: At the end of the Introduction, we added a justification for adding both studies to the same manuscript. While the studies did not inform each other, we considered that both could provide a comprehensive picture of the experiences of families living in areas with a similar deprivation profile and inform policies and practices affecting these families. 

Reviewer #1: Study 1

Uneven gender distribution that should be reflected in your discussion and comments.

Response: In the Discussion of the results of Study 1, we added that the study sample had an uneven gender distribution, being mainly constituted by women. We also added that only women reported a Negative impact on parents' emotional and psychological well-being, and that it was not possible to analyse if gender was a predictor of this experience.

Reviewer #1: I would consider adding reflexivity statements to both of your studies rather than just a statement regarding experience in the methodology.

Response: We removed the information on the authors from the Data analysis section and included it in a new section entitled Reflexivity, which we created within the Methodology of each study. In the Reflexivity section, we included information on which author/s conducted the data collection and respective credentials, occupation at the time of the study, gender, experience and training. We also added a reflection on personal and contextual aspects shaping research.

Reviewer #1: line 306 you allude to positive change seen in other research but don’t specify or consider how/why? How does your specific group compare?

Response: In the Discussion of the results of Study 1, we added information on who the participants were and which positive changes were reported in the referenced studies. Content that was originally in the General discussion (addressing both studies) was moved to the Discussion of results of Study 1 as a way of avoiding repeating content.

Reviewer #1: The discussion largely repeated the results with brief comments made regarding the implications. You missed an opportunity to explore these alignment with research and recommendations specific to each study (though I note you include a larger discussion later). My advice is reduce replication of results section and extend implications and considerations in study 1 discussion.

Response: In the Discussion of results of Study 1, we sought to reduce replication of results and extend considerations on previous research and potential implications for policy and practice aiming to support families.

Reviewer #1: Study 2

This is a difficult group to sample and you are commended for your efforts. Clarity earlier in the paper about the differentiation of focus groups by age might be useful as it is covered in the discussion but definitely made my list of questions early.

Response: In the Participants section of Study 2, we added information on the ages of the children who participated in the Focus Group Discussions. 

Reviewer #1: Results

All there groups appear to be presented together. This needs to be discussed and if sub themes are not addressed by all groups this identified. For example, it seems aspects of educational challenges due to device access were not addressed by children (or you don’t have examples of them). Given you have decided to reflect all stakeholders together I suggest you qualify the groups that spoke to each theme and sub theme so we can better appreciate the views of all stakeholders as unique groups not just as a collective.

Response: In the Results section of Study 2, we added information about the themes, sub-themes and groups of participants contributing to each theme and sub-theme. Within themes and sub-themes, we have grouped quotes by participant group, that is children’s quotes together, parents’ quotes together and service providers’ quotes together. 

Reviewer #1: Some of the quotes are overly long and could be synthesised but still be representative. Also in some sections the quotes do not directly address the comments above it.

Response: In the Results section of Study 2, we have shortened the quotes where possible, by removing some words not necessary. However, we were also conscious of trying to ensure the same meaning as conveyed by participants. We have removed quotes we felt did not address the comments above them. We have tried to replace them with quotes relevant to the points. We also added some quotes to better illustrate our findings.

Reviewer #1: 648 I suggest you draw specific attention to technology and WIFI access as clear examples of the divide you speak of

Response: In the General discussion, we added that families in disadvantaged contexts might be less likely to have adequate broadband and hardware, and extended considerations on the potential implications for supporting families.

Reviewer #1: 653 information as you suggest may not be enough without concrete resourcing. I would also note that international studies suggest that thee vulnerable groups are less likely to return to school so some of the ‘impact’ data you speak of is already there to reference.

Response: In the General discussion, we sought to highlight the importance of concrete resources (such as digital tools) to support families experiencing pandemic-related uncertainty, and measures to support vulnerable groups who are less likely to return to school.

Reviewer #1: Overall an interesting paper that really just needs elaboration, specificity and qualification in some areas.

A pleasure to read.

Response: We thank the Reviewer for the constructive and clear feedback. We consider that the comments helped us improve the manuscript’s elaboration, specificity and qualification.

---

## [Editor Report · Decision Letter 1]

13 Jun 2022

Perspectives and experiences of Covid-19: Two Irish studies of families in disadvantaged communities

PONE-D-21-40789R1

Dear Dr. Leitao,

We’re pleased to inform you that your manuscript has been judged scientifically suitable for publication and will be formally accepted for publication once it meets all outstanding technical requirements.

Kind regards,

Gerard Hutchinson, MD

Academic Editor

PLOS ONE
---

## [Editor Report · Acceptance letter]

10 Jul 2022

PONE-D-21-40789R1 

Perspectives and experiences of Covid-19: Two Irish studies of families in disadvantaged communities 

Dear Dr. Leitão:

I'm pleased to inform you that your manuscript has been deemed suitable for publication in PLOS ONE. Congratulations! Your manuscript is now with our production department. 

Kind regards, 

on behalf of

Dr. Gerard Hutchinson 

Academic Editor

PLOS ONE